# Characteristics of Vortices around Forward Swept Wing at Low Speeds/High Angles of Attack

**Masahiro Kanazaki** *,† [ID] **and Nao Setoguchi** †

Department of Aeronautics and Astronautics, Graduate School of Systems Design, Tokyo Metropolitan University, Tokyo 191-0065, Japan; k9644046@kadai.jp
* Correspondence: kana@tmu.ac.jp
† These authors contributed equally to this work.

**Abstract:** The forward-swept wing (FSW), one of the wing planforms used in aircraft, is known for its high performance in reducing wave drag. Additionally, a study has shown that this wing planform can mitigate sonic booms, which pose a significant challenge to achieving supersonic transport (SST). Therefore, FSW is expected to find applications in future SST aircraft owing to aerodynamic advantages at high speeds. However, there is a lack of sufficient knowledge and systematization to improve aerodynamic performance at low speeds and high angles of attack during takeoff and landing. These are crucial for practical implementation. Although the aerodynamic benefits of an FSW in high-speed flight can be harnessed using advanced structural and control technologies, the realization of SST using an FSW is challenging without enhanced research on low-speed aerodynamics. This study explores the practical aerodynamic knowledge of FSWs. We utilized a numerical simulation based on the Navier–Stokes equation and focused on investigating wake vortex phenomena. Our simulation included various wing planforms, including backward-swept wings (BSWs). The results revealed the presence of vortices with lateral axes emanating from the FSW, while longitudinal vortices were observed in the BSW. Based on these results, we developed a theoretical hypothesis for the vortex structure around an FSW.

**Keywords:** supersonic wing; forward-swept wing; low speeds/high angles of attack



## 1. Introduction

Aerodynamic advantages of the forward-step wing (FSW), including reduced wave drag during high-speed cruises and improved sonic boom performance for supersonic transport (SST), have been well established [1]. However, several limitations must be addressed to apply FSW to real aircraft. For example, one of the structural problems that often arises with FSW is divergence. This issue can be resolved by applying the composite material and optimizing its structure [2]. In addition, lateral stability is another issue that needs to be addressed for real application. According to [3], the canard wing can control the rolling and yawing stabilities. Additionally, improving the rolling and yawing stabilities can be achieved by optimizing the dihedral angle of the main wing and the size and shape of the horizontal tail wing. The takeoff and landing performance pose major challenges for the FSW because of its smaller maximum lift compared to that of the BSW. The main aerodynamic challenge is to enhance the low-speed, high-angle-of-attack performance, which is crucial not only during the cruise but also during takeoff and landing. To successfully incorporate the FSW into a real-world civil aircraft, a deeper understanding of its flow structure and design of efficient high-lift devices (HLDs) and controls is crucial. In particular, comprehending low-speed- and high-angle-of-attack characteristics of wing planforms for high-speed aircraft is essential for achieving stable takeoff and landing. The flow structure of an aircraft wing surface directly affects the lift-to-drag ratio, stability, and aerodynamic noise. Previous studies [4] have extensively

investigated the vortex development on the existing wing surfaces, such as the backward-swept wing (BSW). Moreover, the comprehensive literature summarizes the relationship between the mainstream velocity, angle of attack, wing planforms, and their corresponding aerodynamic characteristics. However, research that focuses on the low-speed and high-angle-of-attack aerodynamics and the flow structure of forward-swept wings remains relatively limited.

The vortex structure originating from the leading edge is well documented for conventional BSWs. Additionally, numerous studies have been conducted on HLD designs in this context. In 1966, Razak et al. published NASA Contractor Report (CR)-421 [4], which provided a comprehensive overview of the theoretical and experimental findings for rectangular and backward-swept triangular/trapezoidal planforms. Furthermore, NASA conducted flight tests on an experimental FSW aircraft, X-29, and reported its aerodynamic characteristics [5]. However, the relationship between wing planform and aerodynamics has not been adequately explained, especially in terms of low-speed and high-angle attack conditions. Although the study by Razak et al. included an FSW, the number of cases examined was limited. Thus, the study cannot be considered a comprehensive and systematic organization of the knowledge. Consequently, a systematic understanding of FSW aerodynamics is lacking. As FSW is one of the various wing planforms, studying effective HLDs for FSW remains challenging.

The objective of this study was to systematically establish practical knowledge regarding the flowfield structure and aerodynamics at low speeds and high angles of attack for an FSW. This knowledge will contribute to future research, particularly on HLD design. To this end, we assumed SST with a forward wing, performed calculations using highly resolved Reynolds-averaged Navier–Stokes (RANS) simulations for various planforms, observed the flowfield, and acquired the relevant knowledge. The remainder of this paper is organized as follows:

1. In the first section, we provide an introduction.
2. The second section introduces the wing model used in this study.
3. The third section outlines the methodologies employed, including computational fluid dynamics, computational mesh generation, and computational conditions.
4. In the fourth section, we present and discuss the numerical results, including flow-field visualizations.
5. The theoretical hypothesis is then discussed based on the numerical results.
6. Finally, the fifth section presents the conclusions of this study.

## 2. Wing Models

In this study, we adopted the concept model of the supersonic business jet (SSBJ), currently under consideration by JAXA as the baseline configuration (Figure 1a) [6]. The model comprises three components: the BSW, fuselage, and horizontal and vertical tails. The main wing as a double-tapered design with an area of 35.4 m$^2$, aspect ratio of 2.6, leading-edge sweep angle of 76° for the inboard section, and sweep angle of $\Lambda = 52°$ for the outboard section. The taper ratio was 0.4 for the inboard section and 0.14 for the outboard section. Figure 1b shows the FSW model in which only the outboard wing section is altered from the baseline configuration. Regardless of the planform, the wing profile remained the same as that of the baseline with a sweep angle of $\Lambda = -52°$ (opposite to the 52° defined in Figure 1c. This FSW planform was previously employed in the literature [1]. Its ability to effectively reduce the sonic boom intensity, better than the BSW configuration during the cruise, has been confirmed. The airfoil geometries at the wing tip, the kink, and the root are shown in Figure 1d. This wing has a rounded leading edge, and the leading edge radius at the tip is 0.057% $c$, which at the kink is 0.06% $c$, and at the root is 0.01% $c$. Here, $c$ is the chord length of each cross section. The components apart from the main wings maintained the same geometry as the baseline configuration. This was to isolate the effects of changes in the main wing.

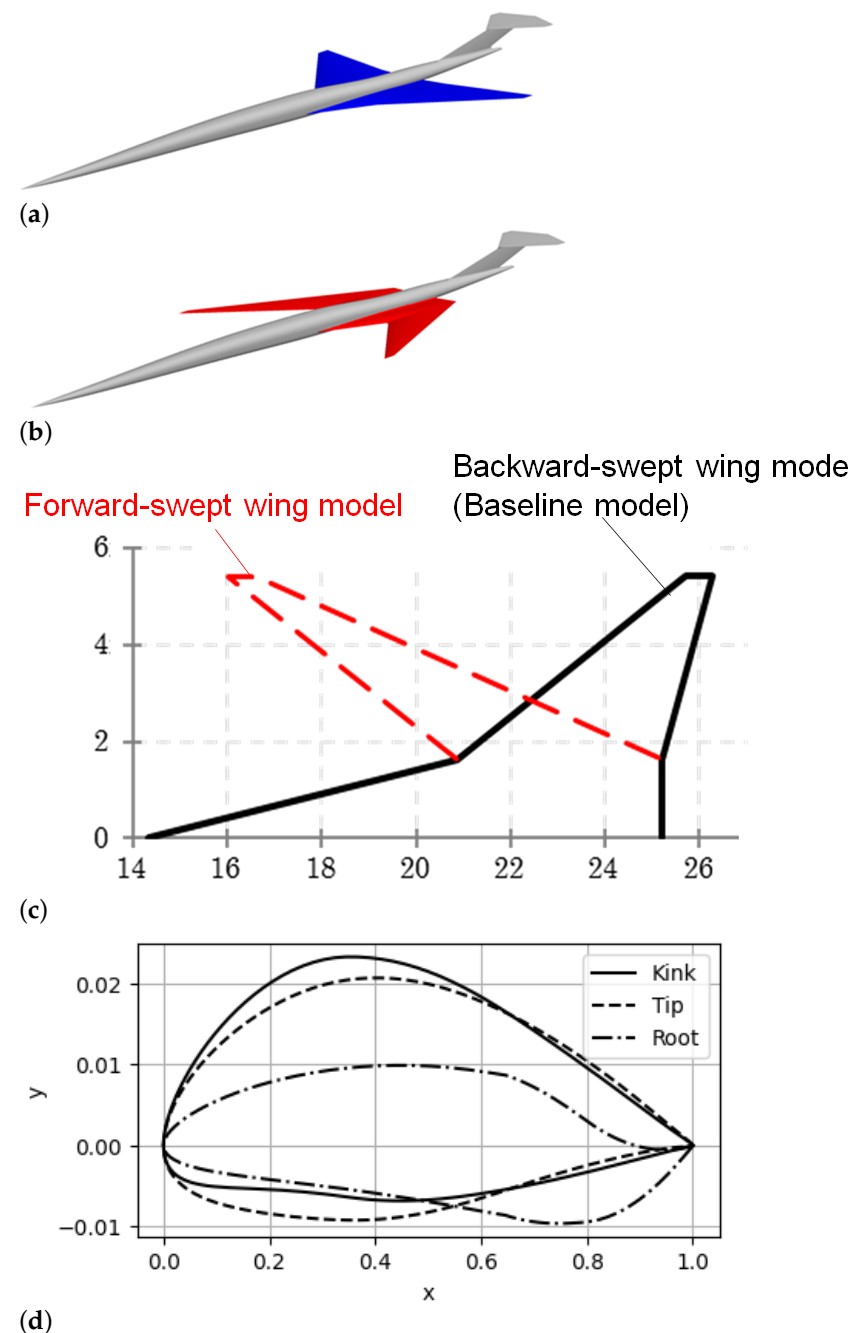

**Figure 1.** Wing planform definitions. (**a**) Model of BSW, (**b**) Model of FSW, (**c**) Wing planform (unit: m), and (**d**) Wing cross-sectional geometries.

## 3. Numerical Simulation

### 3.1. Computational Fluid Dynamics

The CFD solver is based on an unstructured mesh. The governing equations employed were the three-dimensional compressible Navier–Stokes equations, and we conducted RANS simulations. For the numerical flux evaluations, we employed SLAU [7]. The MUSCL method [8] was employed to maintain the second-order spatial accuracy. For time integration, we used the LU-SGS implicit method [9]. For the turbulence model, we employed the Spalart (SA)-noft2-R model [10]. In this study, we utilized a CFD code called FaST Aerodynamic Routine (FaSTAR [11]) developed by JAXA.

In our simulations, we utilized a half model. A computational mesh was constructed using a hybrid unstructured grid consisting of hexahedrons, tetrahedrons, pyramids, and prism elements. The mesh generation process, Richardson extrapolation [12] was

performed to assess the stability of the numerical simulations. A detailed explanation of the Richardson extrapolation can be found in Appendix A. The minimum grid spacing was set to $y^+ = 1$, and the total number of cells in the mesh was approximately 24 million. The computational mesh was generated using HexaGrid [13], which is an automatic grid generator developed by JAXA.

*3.2. Computational Conditions*

To simulate a low-speed regime, such as during takeoff and landing, the Mach number was set to $M = 0.25$, and the flight altitude was $h = 1000$ [m]. The Reynolds number $Re$, based on the body length, was $1.6 \times 10^8$. The investigated angle of attack $\alpha$ ranged from 0 to 45°.

## 4. Results

*4.1. Comparisons of Lift Characteristics*

The lift curve for the BSW, as shown in Figure 2, indicates that the lift coefficient ($C_L$) increased linearly up to $\alpha = 5°$, but beyond that, it increased nonlinearly up to $\alpha = 22.5°$. After reaching its peak value at $\alpha = 22.5°$, the BSW stalled at $\alpha = 25°$. In contrast, the $C_L$ of the FSW, as shown in Figure 2, increased linearly up to approximately $\alpha = 10°$ but increased gradually at higher angles of attack. $C_L$ of the FSW was generally smaller than that of the BSW across most angles of attack. However, unlike the BSW at $\alpha = 25°$, the FSW did not exhibit a sudden loss of lift. Instead, $C_L$ of the FSW remained almost constant after approximately $\alpha = 25°$. These varied characteristics can be attributed to variations in the separation and vortex behavior near the stall conditions, which are discussed in more detail below.

To investigate the lift generation owing to the vortex (vortex lift), we compared the results obtained from both inviscid and viscous calculations. For the inviscid calculation, we employed the potential solver, specifically PanAir. PanAir uses the panel method based on the linear aerodynamic theory to solve the inviscid surface flow [14]. The comparison between the results obtained from the Navier–Stokes solver and the potential solver is presented in Figure 2. Notably, the difference in the lift coefficients between the two solvers was greater for the backward-swept wing (BSW) configuration than for the forward-swept wing (FSW) configuration. These findings suggest that the BSW design has the potential to achieve higher vortex lift compared to the FSW design. The detail discussion for their vortices are discussed in Section 4.2.

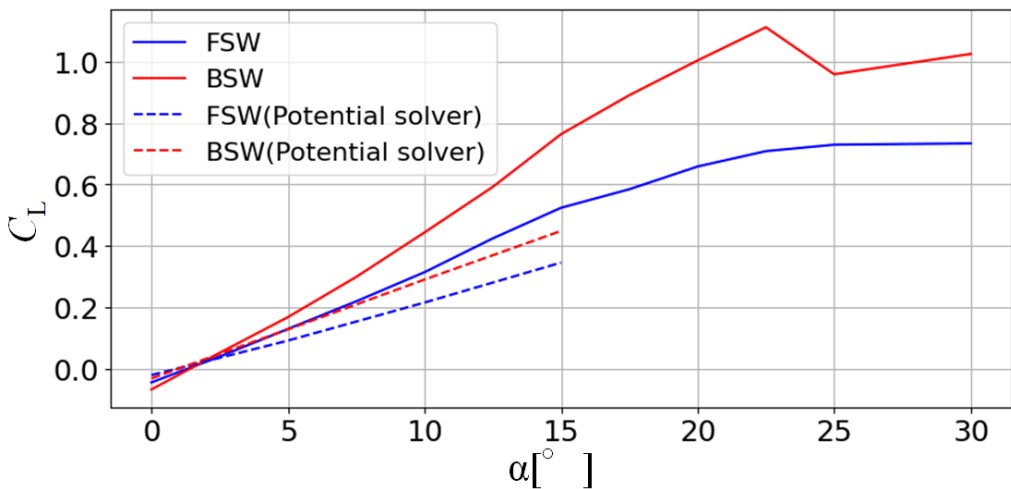

**Figure 2.** Comparisons of the lift curve between BSW and FSW.

*4.2. Comparisons of Separation Vortex Behavior*

Figure 3 illustrates vertex iso surfaces and surfaces pressures on the upper surface of the BSW. At low angles of attack such as $\alpha = 2.5°$ (Figure 3a), the separation vortex

from the leading edge is not clearly observed. However, when $\alpha = 10°$ (Figure 3b), two separation vortices form near the leading edge and generate negative pressure regions. These vortices are referred to as the inboard and outboard vortices. Similar behaviors of separation vortices with a kink has been observed in previous studies [15,16]. At $\alpha = 12.5°$ (Figure 3c), the inboard vortex, which extends in the mainstream direction, moves towards the kink influenced by the outboard vortex. At $\alpha = 15°$ (Figure 3d), the inboard and outboard vortices merge, forming a helical structure. The merged vortex generates a stronger negative pressure on the upper surface than the independent vortices, which is assumed to contribute to the maximum lift slope between $\alpha = 12.5°$ and $15°$. At $\alpha = 22.5°$ (Figure 3e), the merged vortex produces the strongest negative pressure on the upper surface. However, when $\alpha = 25°$ (Figure 3f), the vortex breakdown of the merged vortex occurs on the upper surface, and the strongest negative pressure cannot be observed. Generally, the vortex breakdown point is defined as the tip of an isosurface, at which the mainstream velocity becomes zero [16]. This phenomenon is believed to cause a sudden lift loss at $\alpha = 22°$–$25°$.

Figure 4 shows the flowfield on the upper surface of the FSW. Like the BSW, the inboard and outboard vortices were not clearly observed at low angles of attack, as shown in Figure 4a. However, at $\alpha = 7.5°$ (Figure 4b), the inboard vortex could be confirmed. Unlike the inboard vortex of the BSW, that of the FSW was generated towards the kink because the outboard wing of the FSW was positioned upstream from the inboard wing. Even at small angles of attack, the inboard vortex of the FSW was affected by the outboard vortex. The outboard vortex was clearly observed at $\alpha = 15°$ (Figure 4c). The negative pressure region expanded from the kink to the wingtip along the leading edge. This was due to the washout between the two points, indicating that the leading edge of the kink had a larger effective angle of attack than the wingtip. In addition, at $\alpha = 15°$, the inboard and outboard vortices collided near the kink. Unlike in the BSW, in which the two vortices merged because they had the same rotational direction, the inboard and outboard vortices of the FSW continued to exist individually without merging owing to different rotational directions. This difference in vortex behavior suggests that the lift of the FSW does not decrease sharply but increases linearly. In addition to these vortices, an outboard trailing-edge vortex, not observed in the BSW, was generated at $\alpha = 22.5°$ (Figure 4d). These three vortices continuously expand in the negative-pressure region even after $\alpha = 22.5°$. However, vortex breakdown of the inboard and outboard vortices occurred on the upper surface of the wing at $\alpha = 30°$ (Figure 4e). Subsequently, the negative pressure decreased as vortex breakdown progressed. In contrast, the outboard trailing-edge vortex generated a negative pressure region on the upper surface of the wing as it advanced toward the wingtip. Figure 4e shows that the negative pressure region near the leading edge is negligible. This is attributed to the most rapid breakdown of the inboard vortex near the leading edge compared to the other vortices. As described above, these vortices contribute to the local nonlinear behavior of the lift characteristics of the FSW. At $\alpha = 45°$ (Figure 4f), the wingtip vortex, the outboard vortex, and the outboard trailing-edge vortex approach the wingtip. These vortices remain independent and do not merge, generating negative pressure at the wingtip. This underlying mechanism contributes to the gentler stall characteristics (Figure 2) of the FSW than the BSW. The lifts generated via the outboard vortex and the outboard trailing-edge vortex outweigh the lift loss caused by the breakdown of the inboard vortex at high angles of attack.

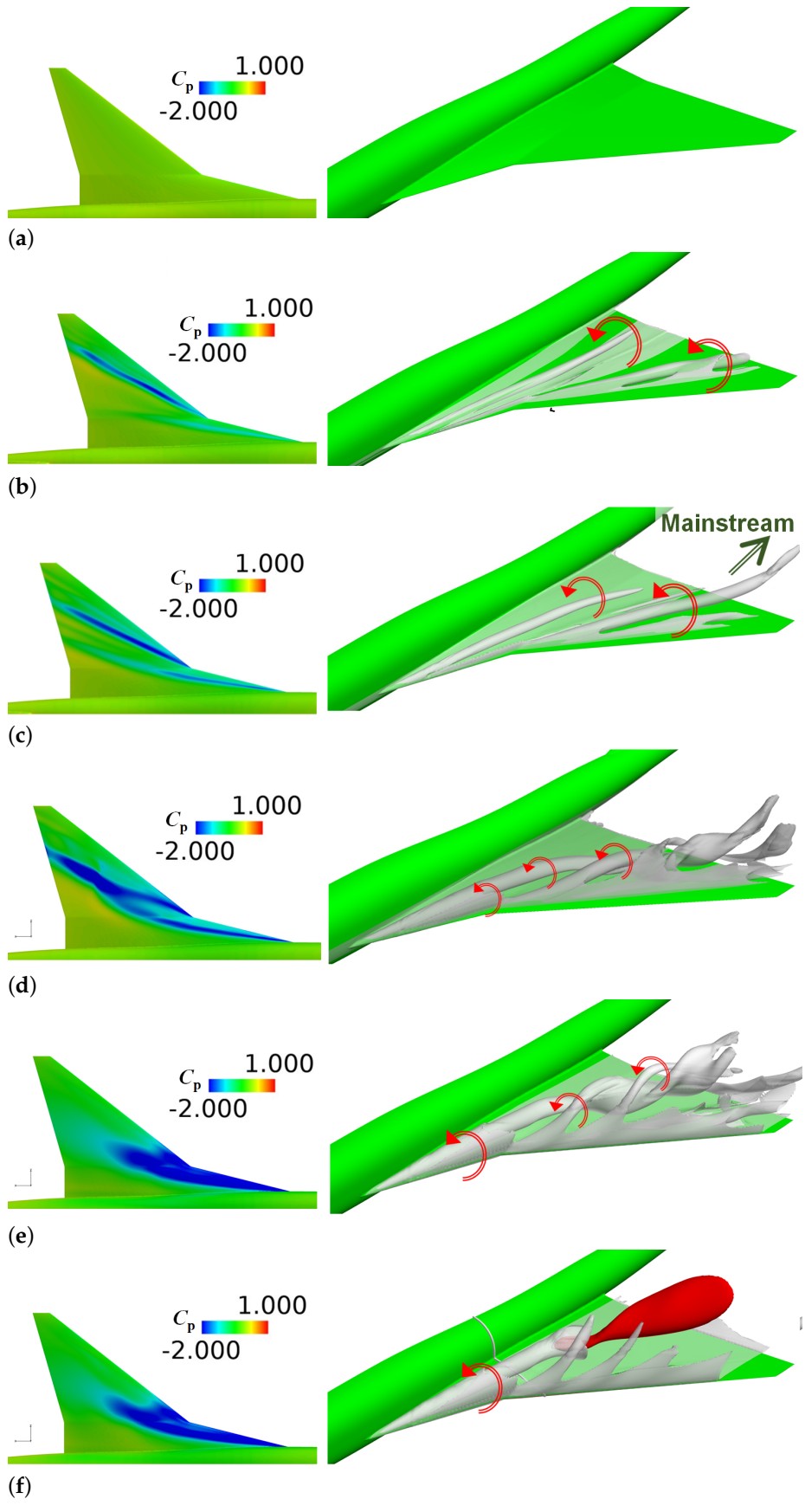

**Figure 3.** Flowfield of the upper surface on the BSW. (Iso-surface of vorticity magnitude indicated gray, velocity in the *x*-direction = 0 represented by red and pressure distribution.) (**a**) $\alpha = 2.5°$, (**b**) $\alpha = 10°$, (**c**) $\alpha = 12.5°$, (**d**) $\alpha = 15°$, (**e**) $\alpha = 22.5°$, and (**f**) $\alpha = 25°$.

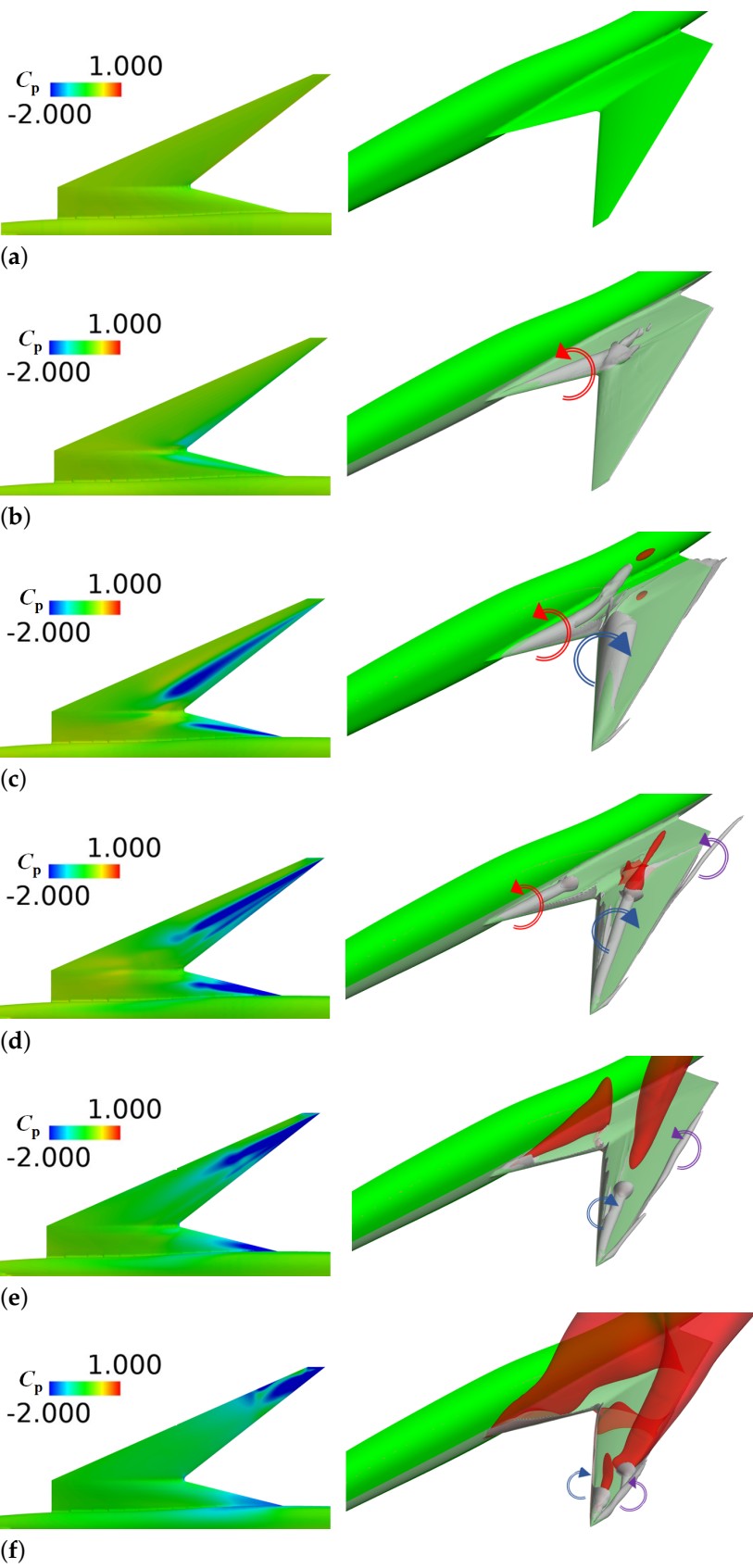

**Figure 4.** Flowfield of the upper surface on the BSW. (Iso-surface of vorticity magnitude indicated gray, velocity in the *x*-direction = 0 represented by red and pressure distribution.) (**a**) $\alpha = 2.5°$, (**b**) $\alpha = 7.5°$, (**c**) $\alpha = 15°$, (**d**) $\alpha = 22.5°$, (**e**) $\alpha = 30°$, and (**f**) $\alpha = 45°$.

### 4.3. Observation of Span-Wise Surface Pressure

Figure 5 illustrates the span-wise pressure coefficients ($C_p$) on the upper surface of the BSW. Figure 5a,b show the $C_p$ distributions at $X = 0.3C_r$ and $X = 0.5C_r$ from the leading edge of the root airfoil, respectively. Here, $X$ is the coordinate along the aircraft longitudinal axis, and $C_r$ (10.1 m) is the root length. In Figure 5a, the suction at 100% semi-span is observed for all $\alpha$, and it is caused by the wing tip vortex breakdown. As the vortex disintegrates at the wing tip, it creates a region of low pressure, resulting in the suction observed at $X = 0.3C_r$ cross section, which is the inboard wing. At 30% local semi-span (Figure 5a), the inboard vortex at $X = 0.3C_r$ cross section continues to amplify even after the breakdown of the merged vortex on the upper surface of the wing. The suction peak at 80% local semi-span corresponds to the inboard primary vortex observed at $X = 0.3C_r$ cross section, whereas the suction peak at approximately 90% local semi-span represents the inboard secondary vortex. The amplification of the inboard vortex contributes to an increase in $C_L$ from $\alpha = 25°$ to $30°$. The four peaks are positioned at $\alpha = 12.5°$ in Figure 5b, which shows the cross sectional $C_p$ at $X = 0.75C_r$ cross section. The first suction peak at 55% local semi-span indicates the inboard primary vortex, whereas the small suction peak at approximately 70% local semi-span represents the inboard secondary vortex at $\alpha = 12.5°$ and $15°$. It is induced by the primary vortex and causes the re-separation of the boundary layer [17]. At $X = 0.75C_r$, the suction peak at approximately 85% local semi-span corresponds to the outboard primary vortex at $\alpha = 12.5°$, exhibiting the strong negative pressure. Thus, compared with the inboard vortex, the outboard vortex contributes more to the lift. The small suction peak at 90% local semi-span is attributed to the outboard secondary vortex at $\alpha = 12.5, 15, 22.5°$. As shown in Figure 3c,d, we can observe that the inboard and outboard vortices approach each other at $\alpha = 12.5°$ and $15°$, and the two suction peaks finally merge into a single suction peak owing to the dominance of the more powerful outboard vortex around $X = 0.75C_r$, which is downstream. During this process, the inboard secondary vortex disappears. For $\alpha = 22.5°$ (Figure 5b), the suction peak representing the maximum negative pressure caused by the merged vortex is observed, which explains the suction peak in $C_L$ at $\alpha = 22.5°$. However, both Figure 5a,b show the disappearance of the suction peaks at $\alpha = 45°$.

Figure 6 shows the span-wise pressure coefficients on the upper surface of the FSW. Figure 6a,b show the $C_p$ distributions at $X = 0.5C_r$ and $X = 0.75C_r$ from the leading edge of the root airfoil, respectively. Comparing Figures 5 and 6, we can observe that the magnitude of negative pressure for the FSW is generally smaller than that of the BSW at various angles of attack. This indicates that the lift of the FSW is not higher than that of the BSW. In Figure 6a, the suction at 100% semi-span at $X = 0.3C_r$ of outboard wing is observed for all $\alpha$, and it is caused by the wing tip vortex breakdown similar to that in BSW cases shown in Figure 5. Additionally, the suction at 84% semi-span at $X = 0.3C_r$ is observed, and it is a result of the normal leading edge suction of the outboard wing after the stagnation point. At $\alpha = 10°$ (Figure 6a), a small suction peak at approximately 87% local semi-span indicates the presence of the outboard vortex at $X = 0.3C_r$. The suction peak corresponding to the outboard vortex becomes more pronounced at approximately 89% local semi-span for $\alpha = 15°$. In addition, the outboard secondary vortex is observed at approximately 87% local semi-span for $\alpha = 20°$ and continues to exist from $\alpha = 20°$ to $40°$, contributing to favorable stall characteristics at $X = 0.3C_r$, which is upstream. Focusing on the outboard primary vortex, the suction peak shifts to the wing tip as the angle of attack increases (Figure 6a) at $X = 0.3C_r$. The maximum suction peak associated with the outboard vortex occurs at $\alpha = 35°$ in Figure 6a; however, it shifts to $\alpha = 20°$ at $X = 0.5C_r$ in Figure 6b. This indicates that the vortex breakdown locus advances and results in a loss of lift as the angle of attack increases. Both the outboard and inboard vortices exhibit a maximum negative pressure at $\alpha = 20°$ for this cross section (Figure 6b). The inboard secondary vortex is observed at approximately 35% local semi-span for $\alpha = 15°$ and $20°$ (Figure 6b). Conversely, at $X = 0.5C_r$, the outboard secondary vortex is observed for $\alpha = 20°$ and $35°$ (Figure 6b). In addition, we can observe a slight resemblance to an

outboard tertiary vortex at $X = 0.5C_r$ at approximately 60% local semi-span. As the most distinctive vortex of the FSW, we can observe a smooth suction peak at approximately 90% local semi-span for $\alpha = 40°$ in Figure 6b. This corresponds to the outboard trailing-edge vortex at $X = 0.3C_r$. Evidently, the behavior of these vortices is closely related to the exceptional stall characteristics exhibited in the FSW.

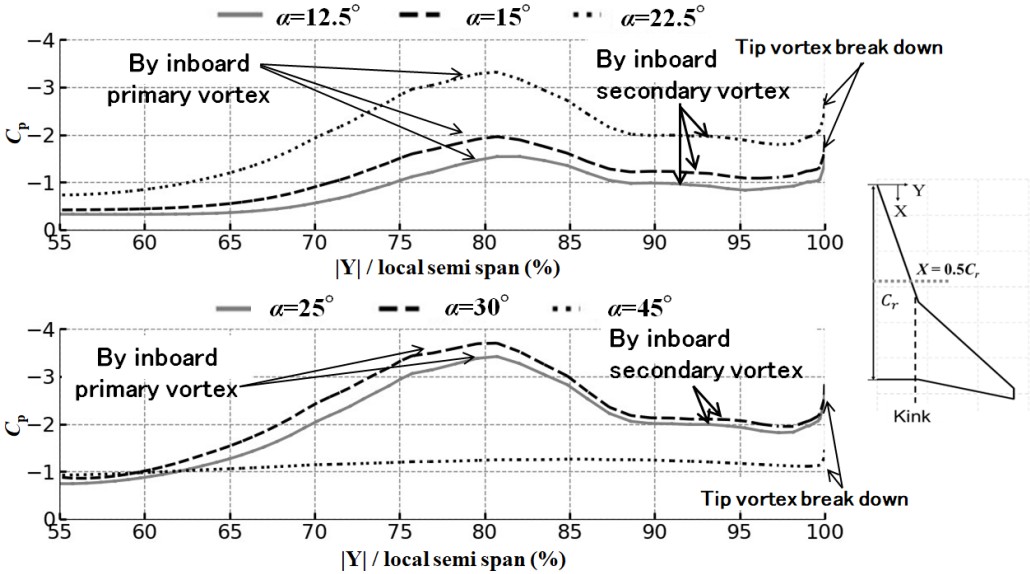

**(a)**

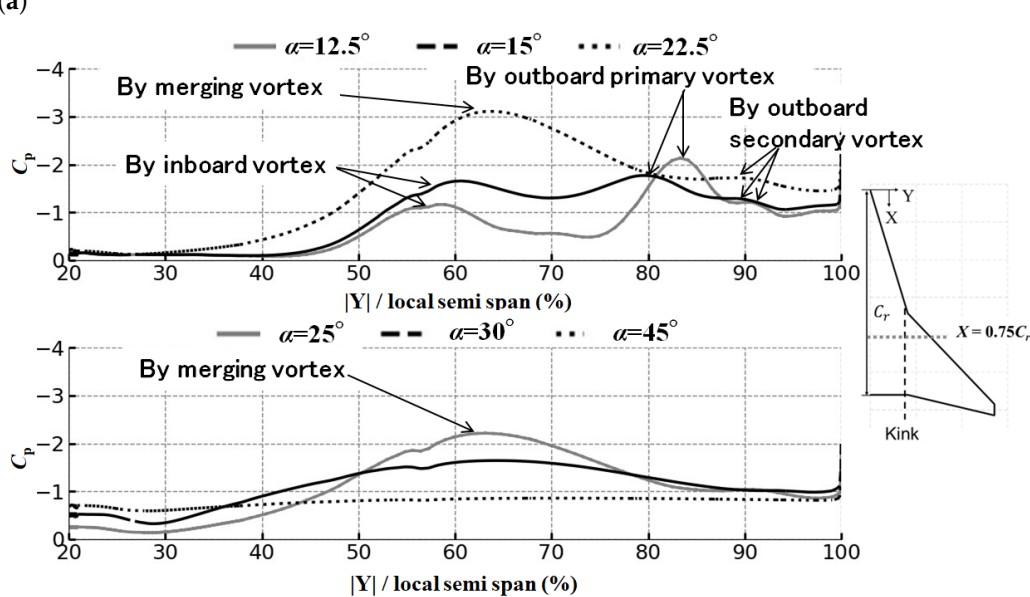

**(b)**

**Figure 5.** Comparison of span-wise pressure coefficients on the upper surface of the BSW. (**a**) BSW along to $C_r = 0.5$ and (**b**) BSW along to $C_r = 0.75$.

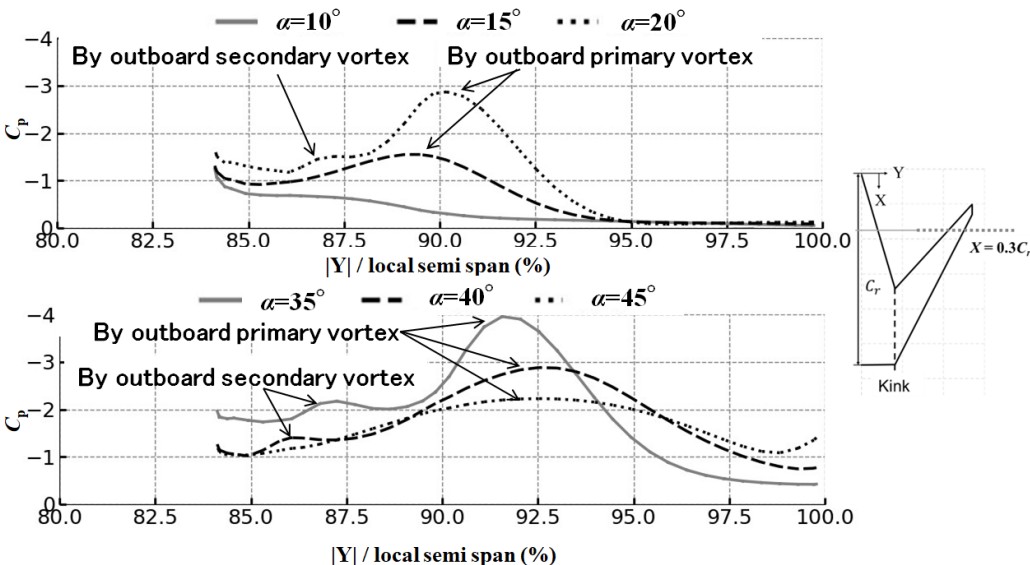

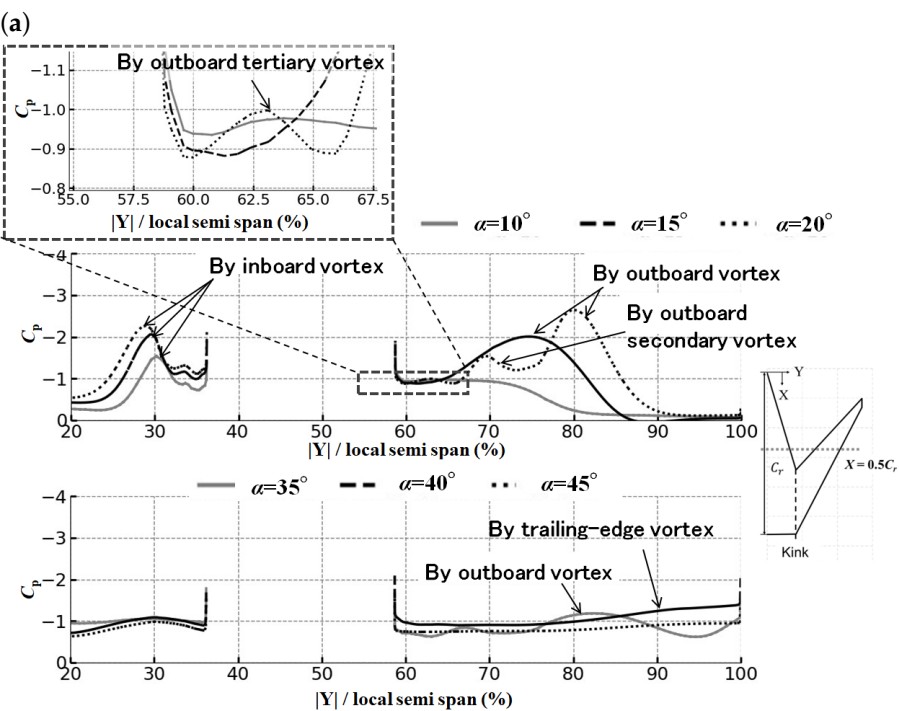

**Figure 6.** Comparison of span-wise pressure coefficients on the upper surface of the FSW. (**a**) FSW along to $C_r = 0.3$ and (**b**) FSW along to $C_r = 0.5$.

### 4.4. Hypothesis of Trailing-Edge Vortex in FSW

The outboard trailing-edge vortex was distinctive and generated in the FSW but not observed in the BSW (Figures 3 and 4). This vortex, shown in Figure 4, is believed to be formed owing to the flow from the lower surface to the upper surface, driven by the pressure difference between them. The vortex is similar to the well-known wingtip vortex. The outboard trailing-edge vortex gained negative pressure via separation at the trailing edge and reattached to the upper surface of the wing (Figure 7a). Additionally, there appeared to be an interaction between the outboard vortex and the outboard trailing-edge vortex, as shown in Figure 4b. The outboard trailing-edge vortex may have been induced by the outboard vortex, and it facilitated the reattachment to the upper surface of the wing. Moreover, as the angle of attack increased, the outboard trailing-edge vortex progressed

along the trailing edge under the influence of the outboard vortex. Consequently, additional negative pressure was generated on the upper surface of the wing, which contributed to the generation of lift.

Based on the numerical results and flow visualization, we propose the following hypotheses (Figure 7b):

1.  A forward-swept wing can be regarded as a delta wing with a yaw angle of incidence, resulting in a flow structure where the leading-edge separation vortices on both sides are biased.
2.  The leading edge of this delta wing with a yaw angle generates the leading edge vortex similar to a normal backward-swept wing. However, a lateral vortex emerges from the leading edge on the trailing side of the main flow, corresponding to the trailing edge of the forward-swept wing, known as the trailing edge vortex.

The authors proposed that this interaction between the trailing-edge vortex and leading-edge vortex governs the observed aerodynamic phenomena.

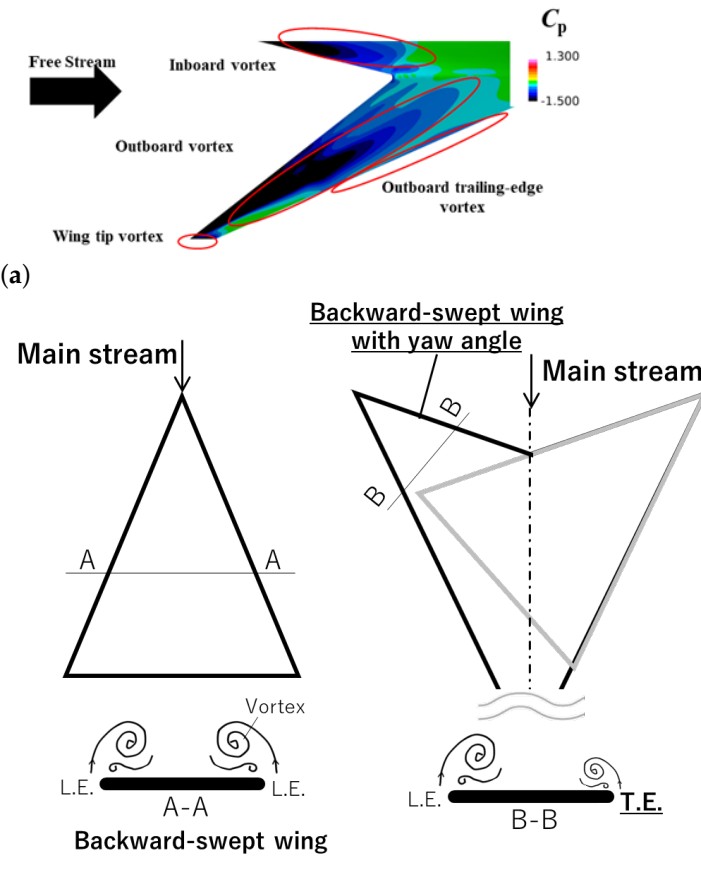

(**a**)

(**b**)

**Figure 7.** (**a**) Pressure coefficient distribution on the upper surface of the FSW, for $\alpha = 30°$ and (**b**) the comparisons of the vortex appearance mechanisms between BSW and FSW.

## 5. Conclusions

To understand the aerodynamic characteristics and separation vortex behaviors in low-speed and high-angle-of-attack regimes for an SSBJ with an FSW, we conducted RANS calculations. The FSW planform used in this study was based on that of the BSW previously studied at JAXA. Changes were made solely to the sweptback angle of the outboard wing. Subsequently, we compared the flowfield characteristics of the FSW and the BSW, including vortex breakdown and aerodynamic properties. Based on our numerical results, the FSW exhibited stall characteristics distinct from those of the BSW. This was due to the

development of negative pressure on the upper surface of the wing by the outboard vortex and outboard trailing-edge vortex, even at high angles of attack. Notably, the appearance of the outboard trailing-edge vortex is a feature of the FSW and plays a significant role in determining its aerodynamic characteristics. As the angle of attack increased, the BSW experienced vortex merging, leading to vortex breakdown at a certain angle of attack. This resulted in a sudden decrease in the lift of the BSW. By contrast, the FSW did not exhibit vortex merging because the outboard and inboard vortices had different rotational directions, allowing them to remain independent. At high angles of attack, an intriguing interaction was observed between the outboard vortex and the outboard trailing-edge vortex. This was an uncommon behavior, and the interaction facilitated the reattachment of the outboard trailing-edge vortex to the upper surface of the wing. This generated a negative pressure at high angles of attack. Consequently, FSW is capable of generating lifts via these vortices at high angles of attack. By performing calculations exclusively for the outboard wing, we discovered that the trailing-edge vortex was generated independent of the wing kink and influenced the aerodynamic coefficients.

In the future, we plan to employ a large eddy simulation (LES) that offers higher fidelity computational fluid dynamics (CFD) than RANS. This will help observe the unsteady vortex phenomena of the FSW. In addition, we planned a wind tunnel experiment to compare the numerical results with the experimental data. Finally, we aim to study the optimal design of HLDs, such as the trailing- and leading-edge flaps, specifically tailored for FSW.

**Author Contributions:** Conceptualization, M.K. and N.S. and M.K.; Formal analysis, N.S.; investigation, M.K. and N.S.; methodology, M.K.; project administration, M.K.; resource, M.K.; software, M.K.; supervisor, M.K.; visualization, M.K. and N.S.; writing—original draft preparation, M.K.; writing—review and editing, M.K. and N.S.; supervision, M.K. All authors have read and agreed to the published version of the manuscript.

**Funding:** This work was supported by JSPS KAKENHI Grant Number 21K04480.

**Data Availability Statement:** Not applicable.

**Conflicts of Interest:** The authors declare no conflict of interest.

## Appendix A. Grid Dependency Study

To assess grid dependence, we conducted a Richardson extrapolation [12] in this study. Three different grids were prepared for BSW: coarse (Figure A1a, approximately 13 million grids), medium (Figure A1b, approximately 24 million grids), and fine (Figure A1c, approximately 56 million grids). All grids were set to y+ = 1. The grids used in the generalized Richardson extrapolation (GRE scheme [12]) scheme are shown in Figure A1d, and the lift coefficients $C_L$ obtained from the CFD simulations are compared across different scenarios. Figure A1 presents the results of the $C_L - (1/N^3)$ grid independence study with GRE at M = 0.25 and an angle of attack $\alpha = 0°$. Medium grids were selected considering the balance between the calculation accuracy and computational time.

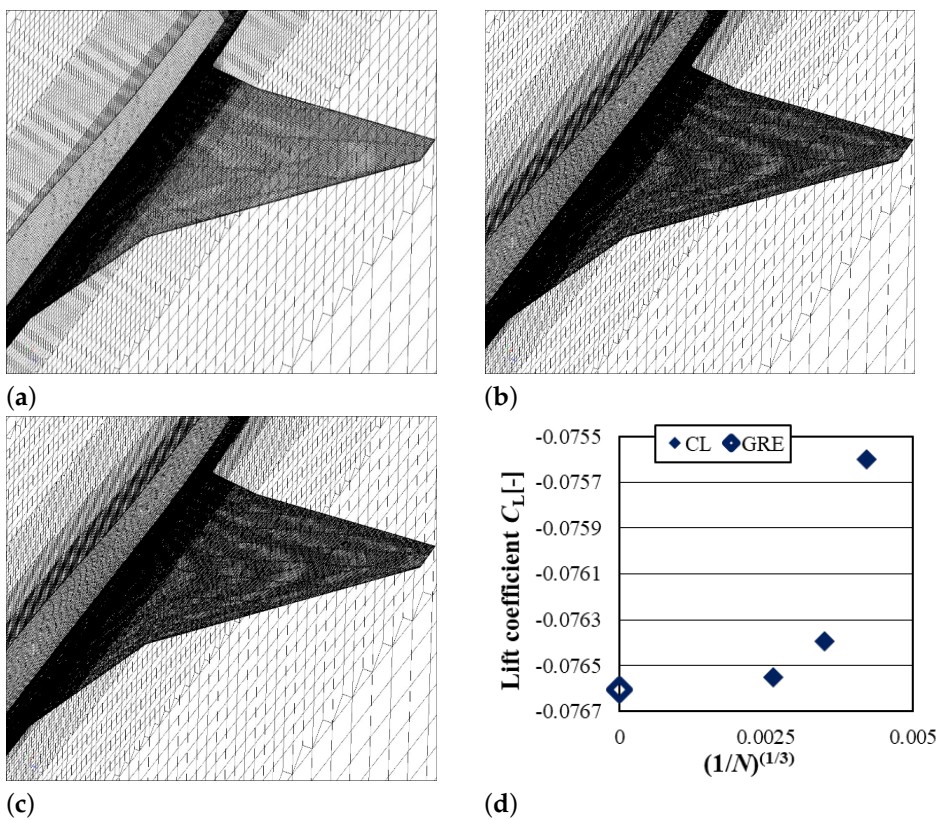

**Figure A1.** Grid dependency study and Richardson extrapolation. (**a**) Coarse mesh, (**b**) Medium mesh, (**c**) Fine mesh and (**d**) Grid convergence of $C_L$.

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
