# Peer review of "Characteristics of Vortices around Forward Swept Wing at Low Speeds/High Angles of Attack"

_aerospace, doi:10.3390/aerospace10090790_

Round 1

Reviewer 1 Report

This paper discusses the aerodynamic effect of forward swept wing upon the SST. The paper indicated different characteristics of the leading-edge separation vortices when compared with those formed on the backward swept wing. The paper also revealed interesting features of the flow. Thus, this paper can be accepted for the Journal when the manuscript is properly revised based on the following points.

a. Title
The reviewer has queer feeling about the words "wake vortex". Since vortices are formed over the forward swept wing, will it be possible to replace "wake vortex" with "leading-edge separation vortices" or just "vortices"? (it is noted that not a single vortex is formed on the FSW)

b. page 1, line 27
"Previous studies": references are necessary.

c. page 1, line 30
"mainline speed": what does it mean? This term is not used in the aerospace fields.

d. chap.1 Introduction
Although the benefit of FSW is mentioned at the beginning of "Introduction", please explain some more details, including the disadvantages of FSW.

e. chap.2 Wing models
Please explain some more details of the wing cross sections such as a name of an airfoil section and airfoil thickness. Does it have a sharp leading-edge or rounded leading-edge? If it has a rounded leading-edge, please specify the leading-edge radius.

f. Figure 2
Will it be possible to analyze the lift curve slope (CLalpha) using the potential flow theory such as vortex lattice theory both for BSW and FSW? If you plot the lift curve slope obtained by this analysis in Fig.2, then the reader may understand the effect of vortical flows upon the CL-alpha curve at low angle of attack range between 0 deg and about 15 deg, because the difference between the CFD results and the above mentioned results indicate the effect of vortical flows formed over the wing.

g. page 4, line 126
"alpha = 25 deg" would be "between alpha = 22.5 deg and 25 deg".

h. page 7, line 194
"approximately 60%": could you explain more in detail about "tertiary vortex"? It is quite difficult to comprehend the flow behavior near 60%, just looking at the figure.

i. Figs. 5 and 6
Please plot the results in color. We cannot distinguish which line corresponds to which angle of attack case.

j. Figs. 5 and 6
Please explain the reason for the sudden Cp decrease at the wing edge such as at 100% of Fig. 5a or 84% of Fig.6a.

k. reference 3
Please specify the details of this reference. "Technical report, 1994." does not mean anything.

Author Response

Dear reviewers
We extend our gratitude to the referees for their meticulous review of our manuscript and for providing valuable
insights. We sincerely apologize for the delay in resubmission. We have diligently revised the manuscript titled ’Characteristics of Wake Vortex around Forward Swept Wing at Low-Speed/High-Angle of Attack’ based on the comments provided by the referees.
Please find attached file that provides responses for your questions and comments.
We look forward to a publication of our manuscript in Aerospace.

Sincerely,
Masahiro Kanazaki

Reviewer 2 Report

The article decribes the numerical investigation of the flow physics about a FSW at low speed and high angle of attack in comparison to a BSW. The emphasis is laid on formation and breakdown of vortices along sharp edges and the impact of these flow phenomena on wing pressure distribution and lift. This topic has rarely been investigated for FSWs and therefore results found here are important to be communicated.

However, for an easier understanding of the complex flow physics, quality of the presentation should be enhanced by improving figures 2, 3, 4, 5 and 6 with an additional revision of section 4.3. "Observation of Spanwise Surface Pressure." Additionally, there are one or two minor clarifications necessary in the remaining text. In particular, the following recommendations are given:

Line 40:  ... especially in terms of low-speed and low-angle attack conditions.
I think this should read: ..... and high-angle of attack conditions. ?

Figure 2: In the line legend the red color denotes the FSW lift curve and blue the BSW but according to the related text it should be the other way around. 

Figures 3 and 4:  Firstly, these important figures should be enlarged at least by a factor of 1.5. Secondly, surface pressure distributions at each alpha should be shown additionally in a top view of the configuration without vortex structures, because in the present figure these conceal cp. The cp color code legend should be bigger (and start at cp = 1.) Visualizing the vortex structure in the 3d images, it would be helpful to somehow indicate the direction of rotation of individual vortices. Vortex breakdown visualization with iso surface of longitudinal velocity = 0 is very good. In Figure 4 the image for alpha = 45 deg is missing (or omitted, 45deg is also missing in the lift curve Fig. 2). 

Figures 5 and 6: In the text it is stated that these figures show spanwise cp-distributions on the upper surface. However, in contrast to his statement, also the lower surface cp's have been plotted. In my opinion, this overloads the graphs. I think, for clarity it would be better to omit the lines for lower surface cp. However, from the line legend it is not possible to distinguish between individual alfa cases. Please improve the line legend. Additionally, in the figure captions you should define parameter Cr. I assume Cr denotes the root chord length, so the cuts you are showing refer to certain x positions as fraction of Cr (as depicted in the inserted small sketches). However, your text does not refer to these x positions at all and therefore it is not an easy task for the reader to understand your analysis. Please improve the discussion in section 4.3 accordingly.

Author Response

Dear reviewers

We extend our gratitude to the referees for their meticulous review of our manuscript and for providing valuable
insights. We sincerely apologize for the delay in resubmission. We have diligently revised the manuscript titled
’Characteristics of Wake Vortex around Forward Swept Wing at Low-Speed/High-Angle of Attack’ based on the
comments provided by the referees. Please see attached file.

We look forward to a publication of our manuscript in Aerospace.

Sincerely,
Masahiro Kanazaki

Round 2

Reviewer 1 Report

The authors have responded properly. Thus this paper can be published as it is.